# Hybrid immunity from severe acute respiratory syndrome coronavirus 2 infection and vaccination in Canadian adults: A cohort study

Patrick E Brown[1], Sze Hang Fu[1], Leslie Newcombe[1], Xuyang Tang[1], Nico Nagelkerke[1], H Chaim Birnboim[1], Aiyush Bansal[1], Karen Colwill[2], Geneviève Mailhot[2], Melanie Delgado-Brand[2], Tulunay Tursun[2], Freda Qi[2], Anne-Claude Gingras[2], Arthur S Slutsky[3], Maria D Pasic[3], Jeffrey Companion[3], Isaac I Bogoch[4], Ed Morawski[5], Teresa Lam[5], Angus Reid[5], Prabhat Jha[1]*, Ab-C Study Collaborators

[1]Centre for Global Health Research, Unity Health Toronto and University of Toronto, Toronto, Canada; [2]Lunenfeld-Tanenbaum Research Institute, Sinai Health, Toronto, Canada; [3]Unity Health Toronto, Toronto, Canada; [4]Toronto General Hospital, University Hospital Network, Toronto, Canada; [5]Angus Reid Institute, Vancouver, Canada

## Abstract

**Background:** Few national-level studies have evaluated the impact of 'hybrid' immunity (vaccination coupled with recovery from infection) from the Omicron variants of the severe acute respiratory syndrome coronavirus 2 (SARS-CoV-2).

**Methods:** From May 2020 to December 2022, we conducted serial assessments (each of ~4000–9000 adults) examining SARS-CoV-2 antibodies within a mostly representative Canadian cohort drawn from a national online polling platform. Adults, most of whom were vaccinated, reported viral test-confirmed infections and mailed self-collected dried blood spots (DBSs) to a central lab. Samples underwent highly sensitive and specific antibody assays to spike and nucleocapsid protein antigens, the latter triggered only by infection. We estimated cumulative SARS-CoV-2 incidence prior to the Omicron period and during the BA.1/1.1 and BA.2/5 waves. We assessed changes in antibody levels and in age-specific active immunity levels.

**Results:** Spike levels were higher in infected than in uninfected adults, regardless of vaccination doses. Among adults vaccinated at least thrice and infected more than 6 months earlier, spike levels fell notably and continuously for the 9-month post-vaccination. In contrast, among adults infected within 6 months, spike levels declined gradually. Declines were similar by sex, age group, and ethnicity. Recent vaccination attenuated declines in spike levels from older infections. In a convenience sample, spike antibody and cellular responses were correlated. Near the end of 2022, about 35% of adults above age 60 had their last vaccine dose more than 6 months ago, and about 25% remained uninfected. The cumulative incidence of SARS-CoV-2 infection rose from 13% (95% confidence interval 11–14%) before omicron to 78% (76–80%) by December 2022, equating to 25 million infected adults cumulatively. However, the coronavirus disease 2019 (COVID-19) weekly death rate during the BA.2/5 waves was less than half of that during the BA.1/1.1 wave, implying a protective role for hybrid immunity.

*For correspondence:
Prabhat.jha@utoronto.ca

**Conclusions:** Strategies to maintain population-level hybrid immunity require up-to-date vaccination coverage, including among those recovering from infection. Population-based, self-collected DBSs are a practicable biological surveillance platform.

**Funding:** Funding was provided by the COVID-19 Immunity Task Force, Canadian Institutes of Health Research, Pfizer Global Medical Grants, and St. Michael's Hospital Foundation. PJ and ACG are funded by the Canada Research Chairs Program.

## Editor's evaluation

This study assessed antibody levels, which are indicative of protection, resulting from both COVID-19 vaccination and natural infection in a representative sample of the Canadian population. The work provides solid evidence that Individuals who received a booster vaccination and had a prior infection had the highest antibody levels, particularly when either the vaccination or natural infection had occurred within the past six months. These findings are of fundamental importance in supporting the value of booster vaccination in populations vulnerable to severe COVID-19.

## Introduction

Infection with the Omicron BA.1/1.1 variant of the SARS-CoV-2 virus occurred worldwide late in 2021 and in early 2022. 'Hybrid' immunity (vaccination coupled with recovery from infection) has emerged as a major determinant of the lower burden of COVID-19 morbidity and mortality in 2022 compared to 2020 or 2021, and as a key determinant of current population-based immunity (*Bobrovitz et al., 2023*; *COVID-19 Forecasting Team, 2023*).

Epidemiological studies have identified hybrid immunity as partially protective against infection or reinfection, and more strongly protective against hospitalization, severe disease, or death (*Bobrovitz et al., 2023*; *COVID-19 Forecasting Team, 2023*; *Altarawneh et al., 2022*). However, such studies rely on the follow-up of hospitalized patients or those with access to polymerase chain reaction (PCR)-based testing, and not randomly selected populations. Thus, the contribution of infection and vaccination to hybrid immunity and the duration of immunity from either exposure remain remarkably poorly documented at the population level (*Tang et al., 2022*; *Brown et al., 2022*; *Centers for Disease Control and Prevention, 2023*; *Goldberg et al., 2022*).

Development of strategies to move from pandemic to endemic management of COVID-19 will be greatly enabled by evidence of population-level immunity, which ideally should be informed by changes over time in biologic measures of immunologic protection (antibody levels, infection status, vaccination, and healthcare utilization). Humoral antibody levels, which correlate strongly with cellular immunity (*Feng et al., 2021*), are the most practical method to monitor populations.

Canada provides an opportunity to document hybrid immunity. Although reaching high levels of vaccination reasonably quickly (by September 2021), Canada experienced a large increase in infections from Omicron from December 2021, even among vaccinated people (*Public Health Agency of Canada, 2023*). Vaccines used in Canada (mostly the mRNA and some adenovirus vaccines) trigger antibody responses to the SARS-CoV-2 spike protein and its receptor-binding domain (RBD), but not to the nucleocapsid protein (N) (*Duarte et al., 2022*). This enables serological distinction of infection from vaccination.

In this study, we estimate cumulative SARS-CoV-2 incidence among Canadian adults in 2020 (*Tang et al., 2022*) and 2021 – prior to the Omicron period – and during two major Omicron waves (BA.1/1.1 and BA.2 and BA.5) in 2022 (*Brown et al., 2022*). We assess declines in active immunity and changes over time in age-specific active immunity levels based on prior infection and concurrent vaccination.

## Methods

From May 2020, the Action to Beat Coronavirus (Ab-C) study conducted six serial assessments of SARS-CoV-2 symptoms (via online surveys) and seropositivity (via antibody testing), with five surveys covering about 4000–9000 adults (*Figure 1*). We recruited adults using the Angus Reid Forum, a nationally representative online polling platform that approximately matches Canada's demographic profile (*Tang et al., 2022*). We obtained informed consent from each participant and excluded any

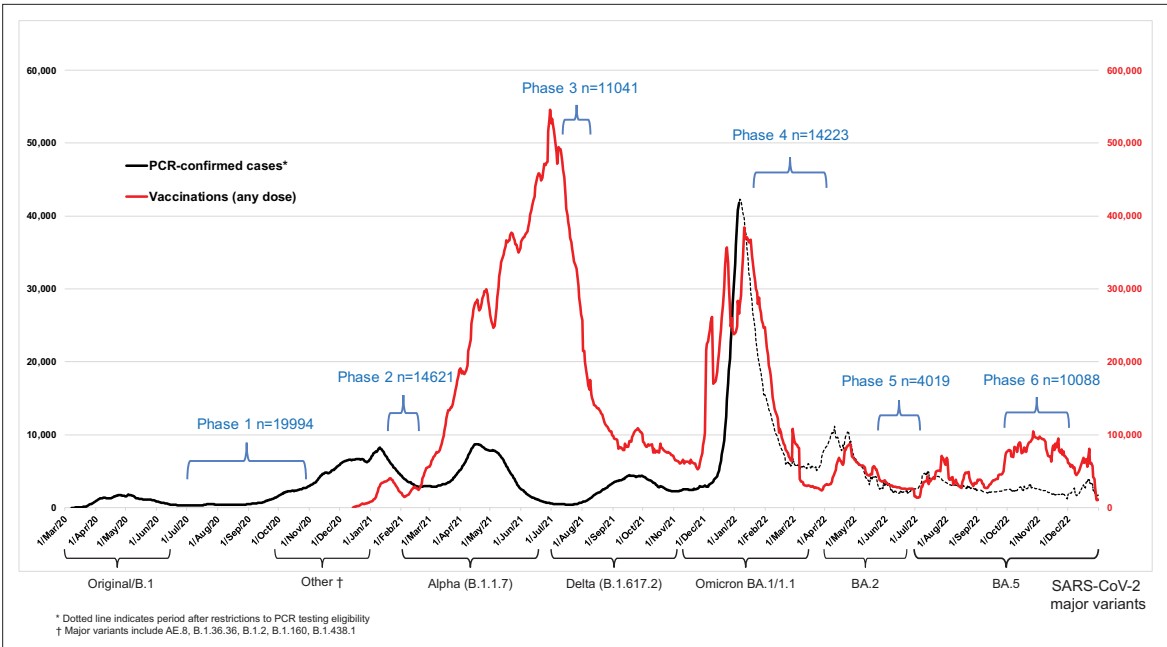

**Figure 1.** Seven-day rolling averages of PCR-confirmed COVID-19 cases in Canada (black solid and dotted line), and SARS-CoV-2 vaccinations (any dose; red line) in relation to the data collection phases of the Action to Beat Coronavirus (Ab-C) study. Testing and vaccination data were derived from COVID-19 Tracker Canada as of 3 February 2023 (https://COVID19Tracker.ca) (**Little, 2023**). Data on major variants were obtained from Public Health Agency of Canada's Health Infobase COVID-19 epidemiology update (https://health-infobase.canada.ca/covid-19/testing-variants.html) (**Public Health Agency of Canada, 2023**). Dotted lines for PCR-based testing after 1 January 2022 reflect the major uncertainty in PCR-based testing. Widespread PCR testing guidelines became stricter and were significantly scaled back in community settings and thus became far less reliable to monitor trends.

unconsented panelist from the study. Details of the sampling, antibody testing strategy, and analyses have been published earlier (**Tang et al., 2022**; **Brown et al., 2022**; **Wu et al., 2020**; **Tang et al., 2021**). The supplementary methods and **Appendix 2—figure 1** report the recruitment, the dried blood spot (DBS) sample return rates, and the few exclusions from the six phases of the study.

The online survey assessed demographic characteristics, history of smoking, hypertension, obesity (self-reported height and weight), diabetes, experience with SARS-CoV-2 infection symptoms, and COVID-19 testing (PCR or rapid antigen). At the end of the survey, respondents indicated their willingness to provide a blood sample by finger prick, and we sent consenters a DBS collection kit with instructions to self-collect. DBS samples were returned to Unity Health laboratories in Toronto, with mail transit times ranging 3–6 days. Sinai Health in Toronto conducted highly sensitive and specific chemiluminescence-based enzyme-linked immunosorbent assays targeting the spike protein, RBD, and N; validation of the assays is reported elsewhere (**Colwill et al., 2022**; **Isho et al., 2020**). Various quality control steps focussed on reducing false positives and false negatives, as well as adjusting the dilution to better detect antibody signals after vaccination became widespread (Appendix 1 provides details of the lab methods and analyses). We conducted cluster analyses of N-positivity (defined below) to assign a probability of seropositivity to each sample using control samples and those with known past viral testing results (**Appendix 2—figure 2**). In a subset of 39 adults in Toronto selected conveniently, we collected venous blood samples at home, and tested these centrally for cellular immunity using the Euroimmun Interferon Gamma Release Assay (**Fernández-González et al., 2022**) to detect T-cell activity against the spike protein (supplementary methods).

Our primary outcomes were the relative levels of antibodies to the spike protein (hereafter 'spike levels'), which are increased both by vaccination and infection (defined as N-positivity or self-reported PCR/rapid test positivity), as a proxy for hybrid immunity levels. Our secondary outcome was the combination of vaccination history and infection. We applied the age-specific cumulative incidence of SARS-CoV-2 to the Statistics Canada national population totals (**Statistics Canada, 2023**) to derive estimates of the number of adults infected in each major phase and compared cumulative incidence to confirmed COVID deaths by phase. Confirmed COVID deaths in Canada (**Public Health Agency of**

*Canada, 2023*) are within 10% of analyses that apply excess all-cause mortality as an upper bound for COVID-19 mortality (*World Health Organization, 2022*).

## Results

We examined three time periods: (1) March 2020 to December 2021 when Canada faced waves of ancestral, Alpha, and Delta variants of SARS-CoV-2; (2) January–March 2022 during the Omicron BA.1/1.1 wave; and (3) April–December 2022 during the Omicron BA.2 and BA.5 waves. *Figure 1* provides the timeline for Phases 1–6, in relation to national weekly averages of confirmed COVID-19 cases and weekly averages of vaccination from any dose.

We surveyed 10,088 adults in Phase 6 of Ab-C, of whom 4025 provided DBS from 26 September to 21 November 2022, and of whom 3378 provided both surveys and DBS. Study participants were comparable to Canadian adults in prevalence of obesity, smoking, diabetes, and vaccination, but fewer lower-education adults participated (*Appendix 3—table 1*). More females and vaccinated adults provided DBS in Phase 6. Lack of vaccination and lower education were correlated (Appendix 1), so we adjusted cumulative incidence for vaccination status. The characteristics of the cohort changed

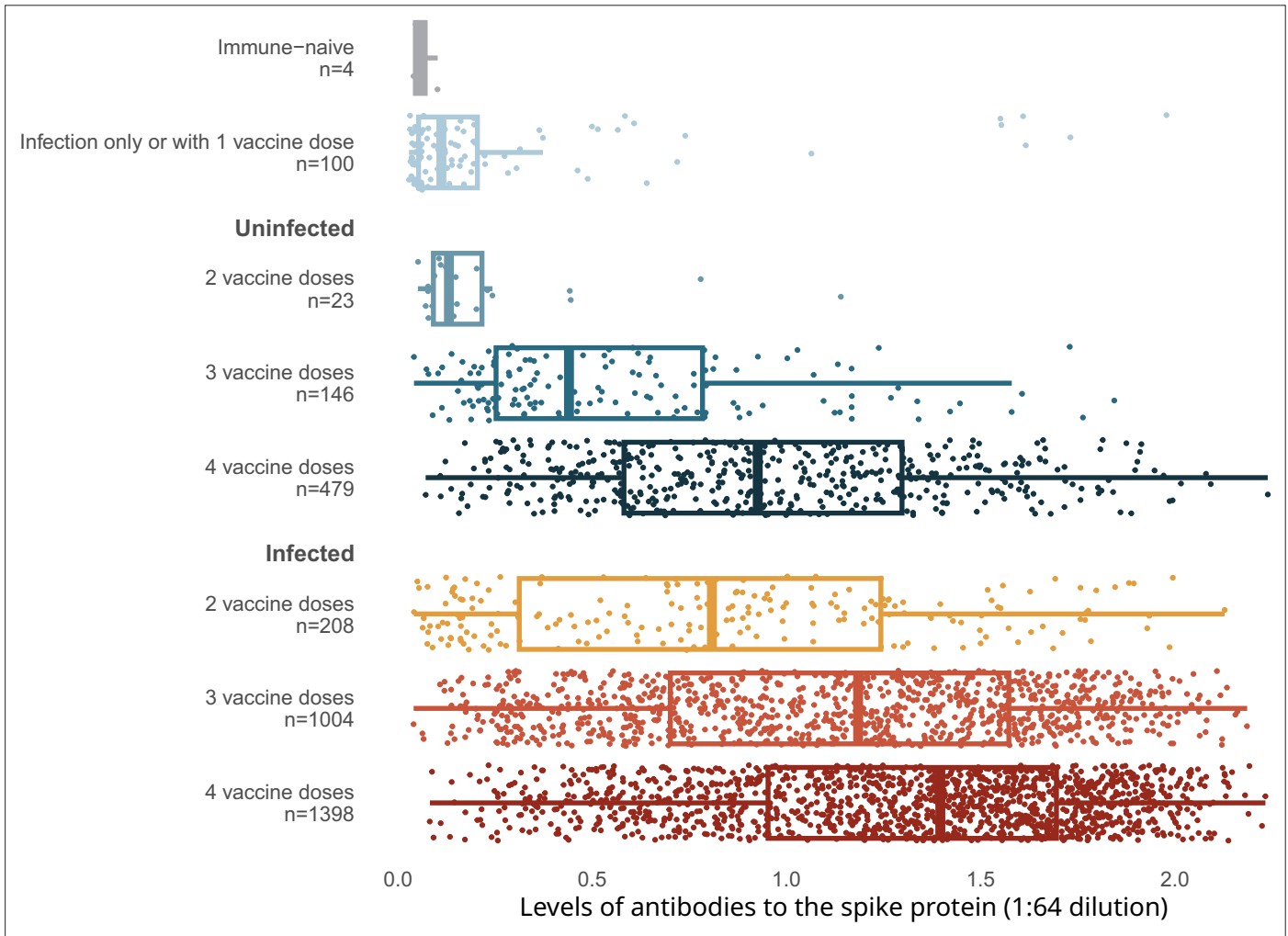

**Figure 2.** Levels of antibodies to the spike protein stratified by infection and number of vaccination doses. Circles represent individuals with their last vaccination (or unvaccinated) >10 days prior to dried blood spot (DBS) sample collection (*n* = 3378 with complete information available as of the time of analyses after excluding 14 low-quality samples). We further excluded 16 participants whose samples were seronegative and viral test was positive, but who did not provide viral test dates or reported test dates less than 8 days from the receipt of DBS. The solid-coloured line represents the median and box plots show the interquartile range. The results above a relative level of 1.2 are outside the linear range of the assay. Results using the receptor-binding domain antigen were similar to the spike protein (*Appendix 2—figure 3*).

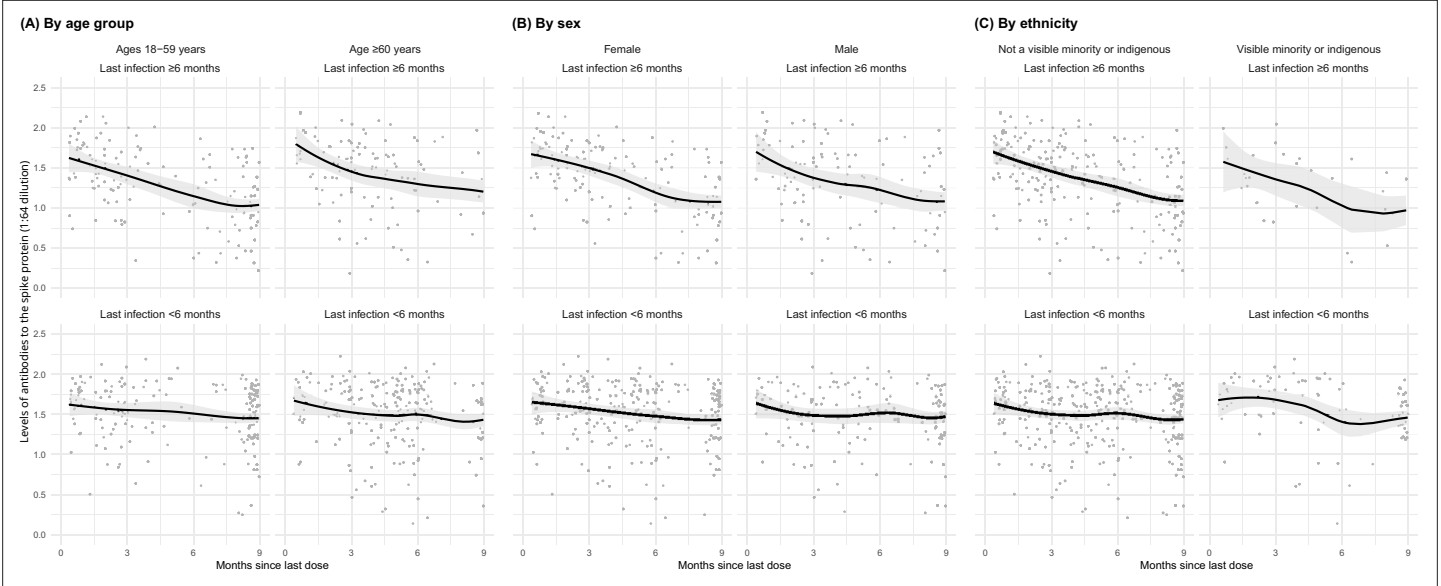

**Figure 3.** Age- (**A**), sex- (**B**), and ethnicity-specific (**C**) trends to 9 months in levels of antibodies to the spike protein among adults vaccinated with three to four doses, stratified by infection more than 6 months ago or less than 6 months ago. See footnote to *Figure 2* for testing details. We created smoothed curves and 95% confidence intervals using locally weighted scatterplot smoothing with span parameter of 0.8 (*Fox and Weisberg, 2018*).

little between Phases 3, 4, and 6 (*Appendix 3—table 1*), so changes in antibody levels are unlikely to be confounded by differential recruitment in each phase (*Tang et al., 2022*; *Brown et al., 2022*).

Canada had four major viral waves before December 2021 and a major increase in vaccination coverage with two doses peaking in early July 2021 (*Figure 1*). A large Omicron BA.1/1.1 wave of January–March 2022 coincided with a large increase in vaccination, mostly of third (booster) doses. The six Ab-C phases captured Canada's major infection and vaccination peaks in a reasonably timely manner.

Spike levels were higher in infected than in uninfected adults, regardless of vaccination doses (*Figure 2*). Spike levels were higher among those who were infected and vaccinated, and lowest among the very few who remained uninfected and unvaccinated, or had only one vaccine dose, or infection without vaccination. Uninfected adults with four vaccine doses were similar in spike level distribution to infected adults with only two or three vaccine doses. Results using the RBD protein were similar (*Appendix 2—figure 3*).

Among adults vaccinated at least thrice and infected more than 6 months prior to the last vaccine dose, spike levels fell notably and continuously for the 9-month post-vaccination (*Figure 3*). In contrast, among adults infected within 6 months, the decline in spike levels was more gradual. Declines were similar by sex, by age group (15–59 or 60+ years), and among various ethnicities (including visible minorities and Indigenous populations). Vaccination within 6 months boosted spike levels from older infections that would have otherwise fallen, yielding similar spike levels among adults infected more than 6 months ago or infected within 6 months (*Appendix 2—figure 4*). Stratifying by periods of 2 months or less, 3–5 months, and 6 or more months yielded comparable results, albeit with smaller numbers in each stratum (data not shown).

Among a convenience sample of 39 adults, all 32 vaccinated adults had positive spike T-cell responses. The T-cell titers and spike antibody levels correlated (*Appendix 2—figure 5*).

Applying the Ab-C results, after standardizing for vaccination status, region, age, and sex to the national profile of Canadian adult population, yielded estimates of cumulative incidence of SARS-CoV-2 infection rising from about 13% before Omicron to 78% by December 2022. This equates to about 25 million infected adults cumulatively. Canada had about 50,000 COVID deaths from March 2020 to December 2022, corresponding to about 6% higher mortality at all ages versus background death rates (*World Health Organization, 2022*). Over 90% of Canadian COVID deaths occurred above age 60 years (*Public Health Agency of Canada, 2023*). Despite the rising cumulative incidence, the COVID-19 weekly death rate per million population during the Omicron BA.2/5 waves (7.7)

**Table 1.** Cumulative incidence, numbers of infected adults, cumulative deaths, and period COVID-19 mortality rate in Canada during various SARS-CoV-2 viral waves.

| Time period | Cumulative incidence*<br>% (95% CI) | No of adult (age 18 or older) infections in millions | Cumulative no of deaths† | COVID-19 mortality rate per million per week during the relevant period |
|---|---|---|---|---|
| Pre-Omicron 2020–2021 | 12.7 (11.2–14.1) | 3.9 (3.5–4.4) | 30,149 | 8.6 |
| Omicron BA.1/1.1 January–March 2022 | 35.7 (34.0–37.4) | 11.3 (10.7–11.8) | 37,750 | 16.6 |
| Omicron BA.2/5 April–December 2022 | 77.7 (75.7–79.6) | 24.6 (23.9–25.2) | 49,674 | 7.7 |

*Post-stratified for geographic region, age, sex, and vaccination status to derive the mean estimate (supplementary methods).

†We used data by end of December 2021, March 2022, and December 2022 from Public Health Agency of Canada's COVID-19 epidemiology update (https://health-infobase.canada.ca/covid-19/) for total number of deaths (**Public Health Agency of Canada, 2023**). Applying the proportion of long-term care deaths from Long-term Care COVID-19 Tracker (https://ltc-covid19-tracker.ca) to the last period, 19,789 of total cumulative deaths occurred in long-term care. Of all long-term care deaths, about 80% occurred during the pre-Omicron period, mostly during the first viral wave of March–June 2020 (**Figure 1**). Over 90% of all COVID deaths occurred at ages 60 or older.

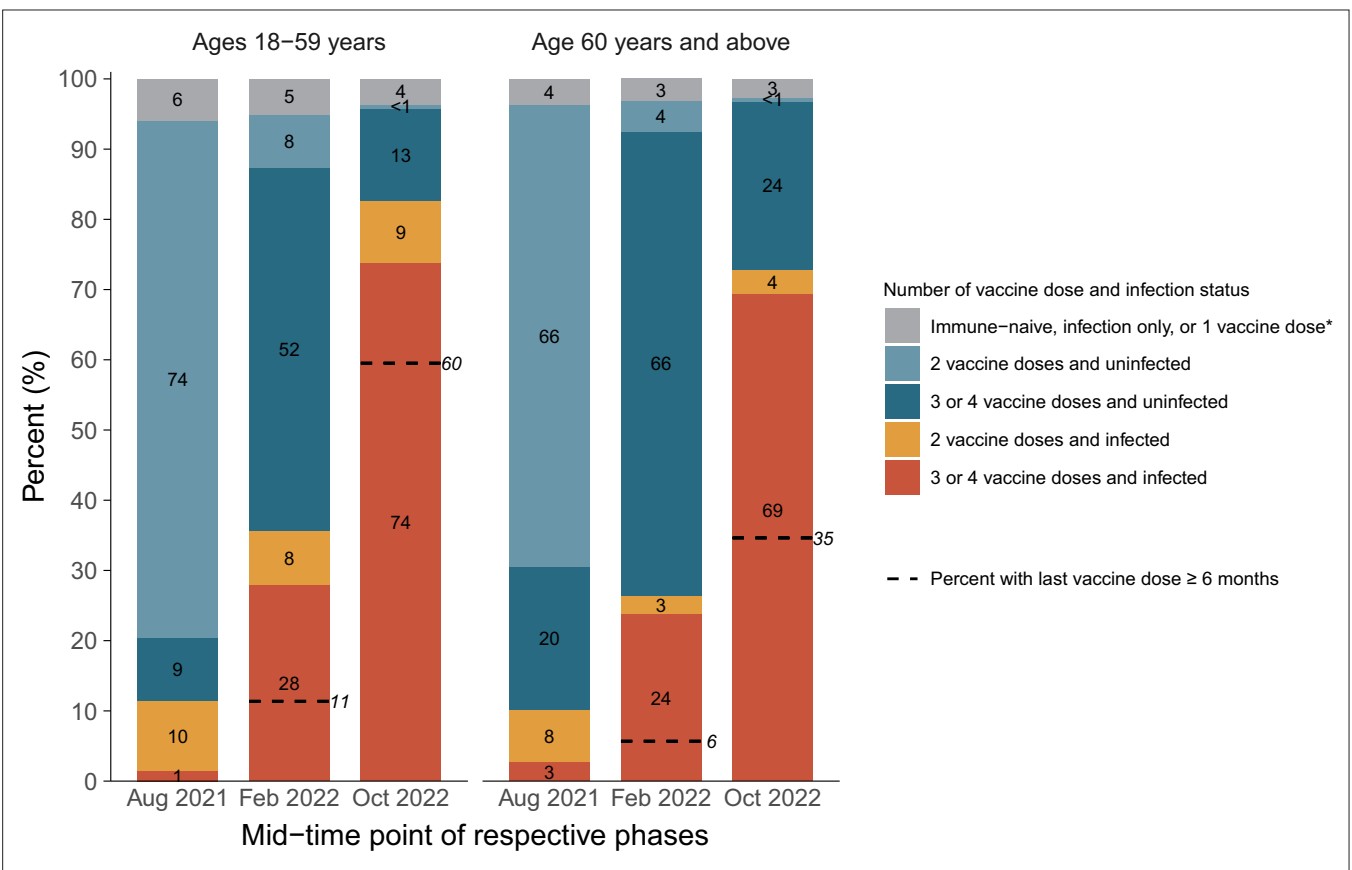

**Figure 4.** Cumulative incidence in each stratum of infection and vaccination in the pre-Omicron wave, during the Omicron BA.1/1.1 wave, and during the BA.2 and BA.5 waves by age group. *Including uninfected and infected cases. The first column in each age group represents the antibody and viral test positivity for the entire period prior to Omicron, whereas the second column represents the values during the Omicron BA.1/1.1 wave and the third during the BA.2/5 waves. By the last time period studied, the numbers of participants aged 15–59 who were N-positive, viral test-positive, and positive to both were 675 (41%), 37 (2%), and 699 (43%). The comparable numbers for participants aged 60 or more were 763 (44%), 35 (2%), and 500 (29%).

was less than half of the weekly death rate during the Omicron BA.1/1.1 wave (16.6). This suggests that hybrid immunity played a role in reducing severe disease and deaths (*Table 1*), at least prior to the eventual waning of the immunity (*Chemaitelly et al., 2022*; *Chemaitelly et al., 2021*).

There were marked increases in infection among younger (18–59 years) and older (60+ years) mostly vaccinated adults, rising from about 11% in each age group by August 2021 to about 86% and 75%, respectively, by December 2022 (*Figure 4*). However, fully 35% of adults above age 60, who are most at risk of hospitalization or death, had their last vaccine dose more than 6 months ago, and about 25% remained uninfected.

## Discussion

We demonstrate the protective nature of hybrid immunity at a population level using robust biological markers of cumulative infection paired with viral testing. While steps to protect individuals and populations from SARS-CoV-2 infection must continue to be implemented, close to 80% of Canadian adults became infected, mostly from the Omicron variants, by December 2022. This high level of infection from the Omicron variants not only led to notable morbidity and mortality, but also contributed to population-level hybrid immunity.

Despite a marked increase in cumulative infection, COVID-19 death rates during Omicron BA.2 and BA.5 were markedly lower than during BA.1/1.1, likely reflecting a strong correlation between protection against severe disease and hybrid immunity (despite lower protection against reinfection). Canadian healthcare systems were overburdened with COVID-related hospitalizations several times during the pandemic. Since summer 2022, hospitalizations have eased significantly, most notably with fewer admissions to intensive care units following the initial Omicron BA.1/1.1 wave (*Public Health Agency of Canada, 2023*). Continued COVID-related practices (most of which were dropped on 1 October 2022), such as travel restrictions, masking mandates, and testing requirements, also may have played a role in the lessened severity of COVID outcomes. Differences in pathogenicity of successive Omicron variants are likely too small (*Strasser et al., 2022*) to explain the differences in COVID-19 death rates.

We showed that absent recent infection, spike levels declined up to 9 months, but reassuringly, declines were comparable in older versus younger adults and by sex and ethnicity. Importantly, recent vaccination attenuated the declines in spike levels from older infections. Obviously, reliance on infections is unwise to boost immunity, especially for those most vulnerable to severe COVID-19. Collectively, our and other studies on hybrid immunity (*Bobrovitz et al., 2023*; *COVID-19 Forecasting Team, 2023*; *Altarawneh et al., 2022*; *Tang et al., 2022*; *Brown et al., 2022*; *Goldberg et al., 2022*; *Tan et al., 2022*) suggest that older adults may require access to booster doses at 6- to 12-month intervals and prior to possible seasonal waves to achieve a robust level of protection against infection. Strategies to maintain population-level hybrid immunity require high vaccination coverage, including among those who have recovered from infection and the few remaining unvaccinated.

The Ab-C study is one of the few nationally representative serosurveys to measure hybrid immunity objectively (*Brown et al., 2022*; *Centers for Disease Control and Prevention, 2023*; *Goldberg et al., 2022*) and has the benefit of sampling the entire population. Large increases from Omicron wave are evident in other Canadian studies (mostly done prior to the BA.5 waves) (*Murphy et al., 2023*). A national US study among blood donors reports lower levels of infection than do we (*Centers for Disease Control and Prevention, 2023*), but has not yet reported on the BA.4/5 waves. Moreover, blood donors or hospitalized patients may have notable biases (*Murphy et al., 2023*). Since the Omicron variant of SARS-CoV-2 appeared, self-testing using rapid antigen tests displaced PCR testing in many countries, including Canada (*Angus Reid, 2022*). The use of spike levels has limitations, although we found it correlated with cellular immunity. Earlier studies demonstrate that high levels of spike or RBD antibodies are predictive of neutralizing antibodies (*Feng et al., 2021*) and correlate with lower viral loads that reduce severe disease in the infected and transmission to others (*Tan et al., 2023*).

Nonetheless, our study has some limitations. First, we had a larger proportion of highly educated adults than the Canadian population. However, the selection biases did not change with subsequent waves, and we saw widespread infection and vaccination in all education groups. We deliberately focussed on distributions of antibody levels which overlap in the comparison categories, but this has the benefit of showing the full range of spike antibody response in the various strata of the infected

and vaccinated. We may be underestimating spike antibody levels due to assay saturation (*Colwill et al., 2022*). N-positivity may have underestimated actual infection because mild cases among vaccinated adults did not mount an antibody response or because people did not seroconvert during the sampling period. Conversely, some adults may have reverted to N-negative status. Finally, defining infection based on cumulative seropositivity and time-specific viral test positivity is crude and made more complicated by periodic viral or vaccination waves. Thus, we are limited in quantifying the hybrid immunity arising from various sequences of variant infections and vaccinations. For example, the apparent plateauing of spike level declines at 9 months in *Figure 3* may reflect cohorts facing at least two distinct vaccination or viral waves. Future phases of our study may assess long-term immunity across different populations, as well as development of long COVID.

Canadian COVID-19 death rates are lower compared to the United States and other similar countries (*Razak et al., 2022*) and we speculate this may be from the sequence of low levels of infection pre-Omicron paired with high vaccination coverage of two doses, followed by a large Omicron wave. Comparative analyses across countries using objective measures of hybrid immunity are required. In Canada and other countries, home-based self-drawn DBSs are a widely practicable and relatively inexpensive monitoring strategy for SARS-CoV-2 population immunity. Despite their limitations, serial serosurveys at the population level are reasonably efficient, low-cost ways to monitor hybrid immunity and to study newer variants of SARS-CoV-2, and possibly even other infectious agents. Future directions could include routine monitoring of various respiratory pathogens, and work to develop practicable multi-plex assays for such infections.

## Acknowledgements

We thank the thousands of Canadians who participated in the Ab-C study. Euroimmun Medical Diagnostics (Sean McFadden) supported the T-cell testing platform at St. Joseph's Health Centre/Unity Health. We thank the thousands of Canadians who participated in the Action to Beat Coronavirus study. A full listing for the Ab-C Study Collaborators is available at https://www.abcstudy.ca. Funding was provided by the COVID-19 Immunity Task Force, Canadian Institutes of Health Research, Pfizer Global Medical Grants, and St. Michael's Hospital Foundation. PJ and ACG are funded by the Canada Research Chairs Program. The funders had no role in study design, data collection, and interpretation, or the decision to submit the work for publication.

## Additional information

### Competing interests

Arthur S Slutsky: Has received consulting fees from Apeiron Biologics, Cellenkos, Diffusion Pharmaceuticals, and GlaxoSmithKline outside the submitted work. Isaac I Bogoch: Has served as a consultant for BlueDot and the National Hockey League Players' Association outside the submitted work. Prabhat Jha: Reviewing editor, *eLife*. The other authors declare that no competing interests exist.

### Funding

| Funder | Grant reference number | Author |
| --- | --- | --- |
| COVID-19 Immunity Task Force | 2021-HQ-000139 | Anne-Claude Gingras<br>Prabhat Jha |
| Canadian Institutes of Health Research | EG2-179433 | Prabhat Jha |
| Pfizer Global Medical Grants | 61608943 | Prabhat Jha |
| St. Michael's Hospital Foundation | | Prabhat Jha |
| Canada Research Chairs Program | | Prabhat Jha<br>Anne-Claude Gingras |

| Funder | Grant reference number | Author |
|--------|------------------------|--------|

The funders had no role in study design, data collection, and interpretation, or the decision to submit the work for publication.

## Author contributions

Patrick E Brown, Formal analysis, Methodology, Writing – original draft, Writing – review and editing; Sze Hang Fu, Formal analysis, Methodology, Writing – original draft, Project administration, Writing – review and editing; Leslie Newcombe, Aiyush Bansal, Ed Morawski, Project administration, Writing – review and editing; Xuyang Tang, Arthur S Slutsky, Isaac I Bogoch, Teresa Lam, Formal analysis, Writing – review and editing; Nico Nagelkerke, Formal analysis, Methodology, Writing – review and editing; H Chaim Birnboim, Karen Colwill, Geneviève Mailhot, Melanie Delgado-Brand, Tulunay Tursun, Freda Qi, Maria D Pasic, Jeffrey Companion, Methodology, Writing – review and editing; Anne-Claude Gingras, Resources, Supervision, Methodology, Writing – review and editing; Angus Reid, Conceptualization, Writing – review and editing; Prabhat Jha, Conceptualization, Formal analysis, Supervision, Methodology, Writing – original draft, Writing – review and editing

## Author ORCIDs

Leslie Newcombe (ID) http://orcid.org/0000-0003-3153-4008
Xuyang Tang (ID) http://orcid.org/0000-0002-2025-3000
Anne-Claude Gingras (ID) http://orcid.org/0000-0002-6090-4437
Prabhat Jha (ID) http://orcid.org/0000-0001-7067-8341

## Ethics

The Ab-C study was approved by the Unity Health Toronto Research Ethics Board (REB # 20-107 and 21-213). All participants provided informed consent to be included in the study.

## Decision letter and Author response

Decision letter https://doi.org/10.7554/eLife.89961.sa1
Author response https://doi.org/10.7554/eLife.89961.sa2

---

# Additional files

## Supplementary files

• MDAR checklist

## Data availability

Ab-C data will be made available publicly through the COVID-19 Immunity Task Force (CITF) Databank. To access the data, please create an account on the CITF Databank portal and submit an application to use the data. Your application will be reviewed by the CITF Databank team. The data access procedure is described in detail at https://www.covid19immunitytaskforce.ca/wp-content/uploads/2022/11/data-access-diagram-en.pdf. This process is free of charge. Analytical code will be available on request in accordance with the Ab-C study's data governance plan. Please email the corresponding author, Dr. Jha at prabhat.jha@utoronto.ca to request the code. The CITF data team harmonizes data from multiple studies funded by CITF, including the Ab-C study. As a result, variable names and labels may change after the harmonization. To minimize confusion when using the code, it's best to have some contact with us when using the harmonized data.

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

## Appendix 1

### Methods

#### Subject recruitment

The Action to Beat Coronavirus (Ab-C) study received ethical approval from Unity Health Toronto (REB 20-107). In Phase 1, from May through September 2020, we invited 44,270 members (out of about 78 000 total members) of the Angus Reid Forum (*Angus Reid Institute, 2024*) an established nationwide polling panel of Canadian adults aged 18 and older, to complete an online survey about SARS-CoV-2 symptoms and testing histories. The sampled population was stratified by age groups (18–34, 35–54, 55+); sex (male, female); education (high school education or lower, some college or college or technical degree, some university, or university degree); and region, by census metropolitan area to match the national demographic profile, with oversampling of adults 60 years or older. In August 2021, we invited about 3100 additional Forum panel members from 17 regions with high burden of infection (of 93 total regions nationwide), based on a regression analysis of SARS-CoV-2 case counts (*Tang et al., 2022*). From December 2020 through January 2021, we invited all 19,994 Phase 1 participants to join Phase 2, retaining the same sampling frame. Phase 3 and 4 recruitment used similar approaches. In Phase 4, we conducted additional outreach to 2587 additional members from marginalized groups at higher risk of SARS-CoV-2 infection (2045 visible minorities and 542 Indigenous individuals). Of these, 1229 agreed to provide DBS and were included in Phase 4 mailouts (919 visible minorities and 310 Indigenous individuals). In Phase 5, a subset of 1304 participants who had recently tested negative for antibodies to nucleocapsid (N) were selected for a supplementary DBS sample; in Phase 6, 5703 DBS participants from any previous phase were enrolled.

Participants were not compensated financially by the study for participating, but earned modest redeemable points from the Angus Reid Forum (*Action to Beat Coronavirus Study, 2021*). Appendix 2 illustrates the study recruitment and flow; there were few (about 1%) exclusions, mostly from incomplete testing.

#### IgG serology

Participants collected five small circles of blood on special bar-coded filter paper, dried the sample for at least 2 hours, placed it in a two-layer protective pouch, and returned it to St. Michael's Hospital in Toronto, postage prepaid. Mailing time across Canada ranged from about 3–8 days. Upon arrival, samples were scanned, catalogued, and stored at 4°C in larger boxes with additional desiccant, and monitored for humidity levels (kept <20%).

Antibodies were then eluted from a 4.7-mm punch in 99 µl of phosphate buffered saline (PBS) + 0.1% Tween (PBS-T) and 1% Triton X-100. The use of 99 µl was to ensure sufficient eluate to test three antigens (spike protein, receptor-binding domain [RBD] of the spike, and nucleocapsid protein [N]). Punches were incubated in elution buffer for a minimum of 4 hr with gentle shaking (150 RPM) at room temperature or overnight at 4°C. The samples were then centrifuged at $1000 \times g$ for 30 s.

The Network Biology Collaborative Centre at Sinai Health, Toronto, conducted a high-throughput, highly sensitive chemiluminescence-based enzyme-linked immunosorbent assay (ELISA) targeting the spike protein, RBD, and N. Chemiluminescent ELISA assays were performed as previously described on a Thermo Fisher Scientific F7 robotic platform (*Colwill et al., 2022*; *Isho et al., 2020*) with a few modifications. Briefly, LUMITRAC 600 high-binding white polystyrene 384-well microplates (Greiner Bio-One #781074, VWR #82051-268) were pre-coated overnight with 10 µl/well of antigen (50 ng spike (SmT1), 20 ng RBD and 7 ng nucleocapsid, all supplied by the National Research Council of Canada [NRC]). After washing (all washes were four times with 100 µl PBS-T), wells were blocked for 1 hr in 80 µl 5% Blocker BLOTTO (Thermo Fisher Scientific, #37530) and then washed. 10 µl of sample (2.5 or 0.156 µl of DBS eluate diluted in 1% final Blocker BLOTTO in PBS-T) was added to each well and incubated for 2 hr at room temperature. After washing, 10 µl of a human anti-IgG fused to HRP (IgG#5, supplied by NRC, final of 0.9 ng/well) diluted in 1% final Blocker BLOTTO in PBS-T was added to each well followed by a 1-hr incubation at room temperature. After four washes, 10 µl of SuperSignal ELISA pico chemiluminescent substrate (diluted 1:4 in MilliQ distilled $H_2O$) was added to each well and incubated for 5–8 min at room temperature. Chemiluminescence was read on an EnVision (Perkin Elmer) plate reader at 100 ms/well using an ultra-sensitive detector.

Each 384-well assay plate included replicates of a standard reference curve of a human anti-spike IgG antibody (VHH72-Fc supplied by NRC) (*Colwill et al., 2022*) or an anti-nucleocapsid

IgG antibody (Genscript, #A02039), positive and negative master mixes of pooled serum samples, human IgG negative control (Sigma, #I4506), and blanks as controls. Negative and/or positive DBS controls (defined using plasma serology results) were included in runs in each phase.

For each antigen, raw values (counts per second) were normalized to a blank-subtracted point in the linear range of the standard reference curve to create a relative ratio (hereinafter referred to as antibody levels). The samples were processed at a 1:4 dilution of the DBS eluate (2.5 μl/well of sample) and 1:64 dilution. We used the former to derive positivity threshold and the latter to display antibody level distributions.

## Determining positivity

There is uncertainty in the measured values of the antibodies to N. We sought to reflect this uncertainty in the confidence intervals for prevalence estimates. We used control samples and known positives to estimate the probability of seropositivity for each sample, and we used multiple imputation to account for the unknown true seropositivity status. We estimated log relative rates in a model adjusting for age, sex, region, and vaccination status. Using post-stratification, we computed estimates and confidence intervals for prevalence in the population and various subgroups, adjusting for the representativeness of the sample.

*Appendix 2—figure 2A* shows the histogram of logged N-positivity for known laboratory negative control samples within each testing plate (in blue) and antibody levels from known positive samples from Phase 4. Known positives are individuals who reported a positive COVID-19 test result more than 7 days before their DBS was received. We used maximum likelihood estimation to define skew-normal densities for the case and control samples, shown as solid lines. *Appendix 2—figure 2B-D* shows histograms of observed antibody levels for each phase, along with a fitted density estimated as a mixture of the red and blue densities from *Appendix 2—figure 2A*. We estimated a mixing proportion for each phase (by maximum likelihood), the densities for each component are shown in blue and red for the seronegative and seropositive components, respectively.

For each sample, we calculated a probability of seropositivity using Bayes rule. This probability depends on the mixing proportion as well as the red and blue densities, as when prevalence is high the threshold should be lowered to reduce false negatives. These probabilities are used for multiple imputation, generating 100 datasets where each sample is designated as seropositive or seronegative. For grouping subjects as infected and uninfected in the 'immunity wall' figures, cutoffs for each phase (shown in *Appendix 2—figure 2*) are set so that the expected number of false positives and false negatives is identical.

Prevalence estimates and their confidence intervals use post-stratification, adjusting the study sample to reflect the Canadian distribution of population by age, sex, region and vaccination status. For each phase, we fit a logistic regression model which included vaccination status (no doses v. one or more) and region (British Columbia and Yukon; Prairie provinces, NWT, Nunavut; Ontario; Quebec; Atlantic provinces), each of which are interacted with age and sex (and the age–sex interaction). We did not interact vaccination status with region, as the number of unvaccinated individuals in the sample was small in some regions. We obtained estimates of log relative rates and the accompanying variance matrix for each of the 100 imputed datasets and combined them according to Rubin's rule.

The population by age, sex, province, and vaccination status at each phase are obtained from the Public Health Agency of Canada's Infobase. (*Public Health Agency of Canada, 2023*). Weights are calculated for each age–sex–region–vaccination group and a weighted average of group-level prevalences is computed with standard errors obtained from the delta method (*Jackson, 2023*).

## Interferon-gamma release assay T-cell analysis

We selected a convenience sample of adults in the Ab-C study within urban Toronto. After obtaining consent for re-contact, participants attended either a Unity Health Toronto hospital visit or agreed to a home visit. A phlebotomist collected one tube of venous blood from each participant using 7 ml lithium-heparin blood collection tubes. Blood collection tubes were mixed by inversion, stored at room temperature, and delivered to the St. Joseph's Health Centre laboratory within 16 hr of collection to be refrigerated at 2–8°C.

Prior to stimulation, samples were removed from refrigeration for 30 min. For each whole-blood sample, one stimulation tube set from the Quan-T-Cell SARS-CoV-2 kit (EUROIMMUN, ET 2606-3003) was warmed to room temperature. Each set consisted of three stimulation tubes: (1) CoV-

2 interferon-gamma release assay (IGRA) BLANK: no T-cell stimulation, for determination of the individual IFN-γ background; (2) CoV-2 IGRA TUBE: specific T-cell stimulation using antigens based on the SARS-CoV-2 spike protein; (3) CoV-2 IGRA STIM: unspecific T-cell stimulation by means of a mitogen, for control of the stimulation ability. The blood collection tube was mixed by gentle inversion, then sampled using 1 ml pipets to draw and transfer 500 µl of whole blood to each of the three tubes. The filled stimulation tubes were sealed and mixed by rapid inversion, then shaken by hand and incubated at 37°C for 20–24 hr. At the end of the incubation period, the tubes were removed from the incubator and centrifuged for 10 min between 6000 and 12,000 × $g$.

Following centrifugation, the plasma obtained from the stimulated whole-blood samples was diluted and used on the anti-IFN-γ-coated ELISA plate. EUROIMMUN Mississauga conducted interferon-gamma release assays using the Quan-T-Cell ELISA (EQ 6841-9601). 100 µl of the calibrators, controls, and diluted plasma samples (1:5 in sample buffer) were transferred into the individual microplate wells and incubated for 120 min at room temperature. The wells were washed (five times, each using 300 µl of wash buffer). 100 µl of biotin was pipetted into each well and incubated for 30 min at room temperature. The wells were washed, and 100 µl of enzyme conjugate was pipetted into each well and incubated for 30 min at room temperature. The wells were washed, and 100 µl of chromogen/substrate solution was pipetted into each well and incubated for 20 min at room temperature, protected from direct sunlight. 100 µl of stop solution was pipetted into each well. Photometric measurements of the colour intensity were made at a wavelength of 450 nm and a reference wavelength between 620 and 650 nm.

## Epidemiological analyses

This analysis focussed on Phases 3–6 of the Ab-C study, which correspond to the pre-omicron (15 August to 15 October 2021) and omicron (BA.1/1.1, BA.2, and BA.5) periods (24 January to 30 March; 27 May to 1 July; and 26 September to 21 November 2022), respectively. To confirm the Ab-C data are representative of the Canadian population, we calculated the proportion of participants who filled out the survey and provided DBS by demographic characteristics (province, household size, age, sex, education, ethnicity, weight, smoking status, diabetes, hypertension) and vaccination status, and compared these to the Canadian national data (*Appendix 3—table 1*).

As already reported (*Tang et al., 2022*), the demographic and health characteristics of those who completed surveys and provided DBS were generally comparable to the Canadian census population, except for fewer adults with an educational level of some college or less in the Ab-C study compared with the census population. In Phase 6, the proportion of adults unvaccinated was similar in the Ab-C surveyed population (8%) as in Canada overall (10%). However, the unvaccinated rates were lower in those who submitted DBS samples (3%). We have previously found greater unvaccinated rates among the lower levels of education (*Tang et al., 2021*). Education level (some college or less, college graduate, university graduate) was inversely correlated with vaccination status: chi-squared statistic 17.156 (df = 2; p-value of 0.0001882). Hence, we adjusted for vaccination status when calculating estimates of cumulative incidence. Moreover, the Ab-C study has had fewer racial or ethnic minority adults (which is defined by Statistics Canada, the national lead statistical agency, as 'Visible Minorities') but more Indigenous adults than the census population. Compared with the census population or nationally representative surveys, study participants had a similar prevalence of obesity, current or former smoking, diabetes and hypertension.

The Phase 3–6 population distributions, which are most directly relevant to estimating cumulative and period-specific Omicron incidence, are broadly similar among those who completed surveys and those who provided a DBS (*Appendix 3—table 1*) (*Brown et al., 2022*). Finally, a comparison of those invited who participated and did not in Phase 1 of the study showed a bias towards greater female participation (*Tang et al., 2022*). However, differences by sex were not important predictors of cumulative incidence (data not shown), so this bias does not materially affect the overall estimates of cumulative infection.

The age-specific 'immunity wall' in *Figure 4* defines infection as either having tested positive on polymerase chain reaction or antigen rapid test or with antibodies to the N antigen (which is appropriate among the largely vaccinated cohort). N-positivity reflects infection and would not arise from Canadian-approved vaccines that only contain the spike protein. We defined infection as any positive COVID-19 test more than 7 days prior to the DBS being received and any N-positivity.

We obtained the overall cumulative incidence of SARS-CoV-2 infections based on N-positivity and derived the 95% confidence intervals using the delta method (*Goldberg et al., 2022*). In order to examine the level of antibody response from infection and vaccination (by vaccine doses), we display the distributions of antibodies to spike antigens (at the 1:64 dilution) using box plots with jitter (*Figure 2*). Results for antibodies to RBD are similar (*Appendix 2—figure 4*). All analyses were performed using Stata 17 and R 4.2.1.

## Appendix 2

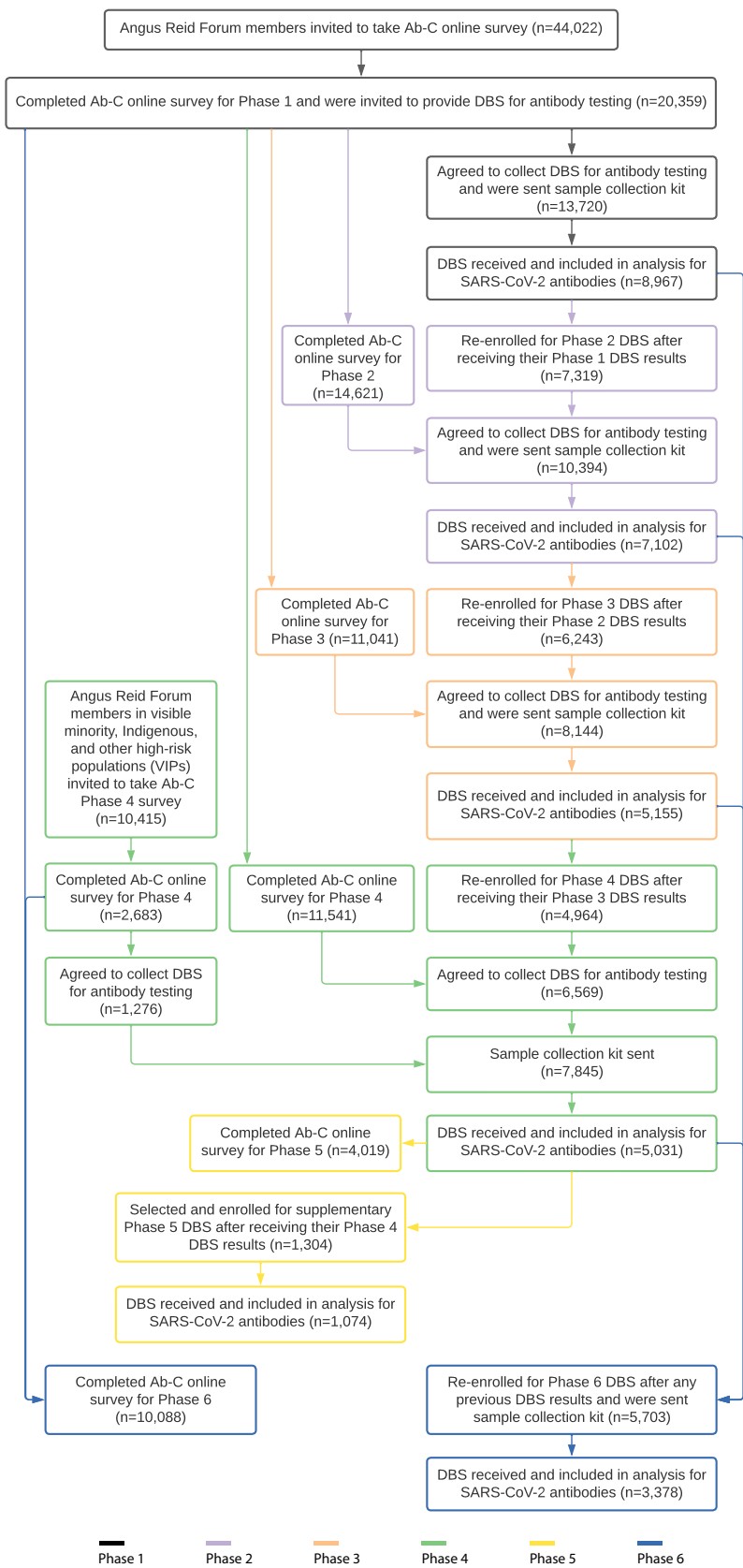

**Appendix 2—figure 1.** Study flow including sampling and study inclusion by phase in the Ab-C study.

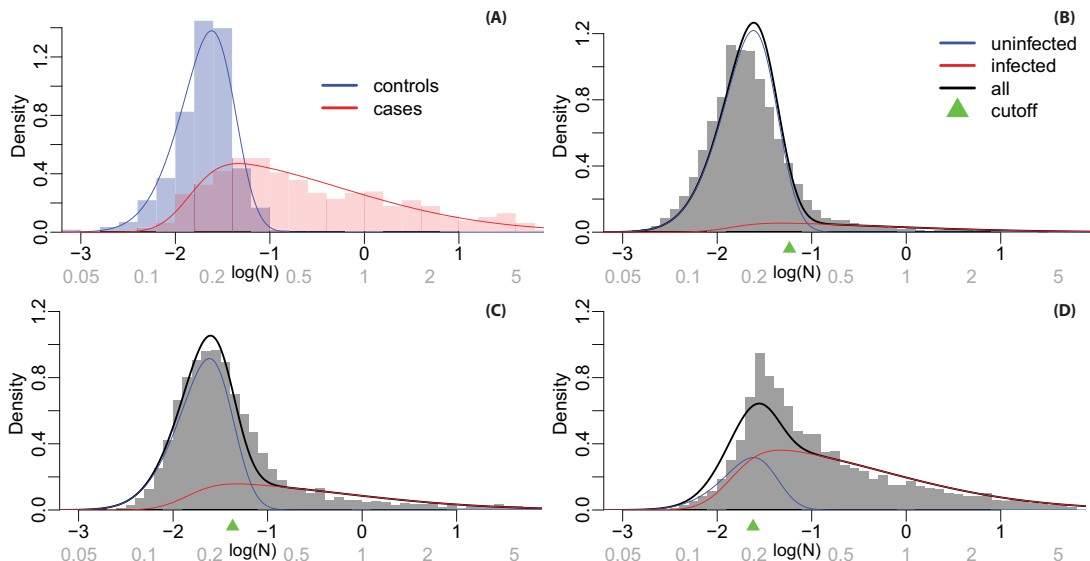

**Appendix 2—figure 2.** Histograms and fitted densities for N-positivity levels. Values on the log scale are shown on the horizontal axis in black and on the natural scale (not logged) are shown in grey. Notes: (**A**) Fitted densities for known cases and controls in calibration samples; (**B**) densities estimated for infected and uninfected individuals from the mixture model in Phase 3 samples; (**C**) Phase 4 samples; and (**D**) Phase 6 samples.

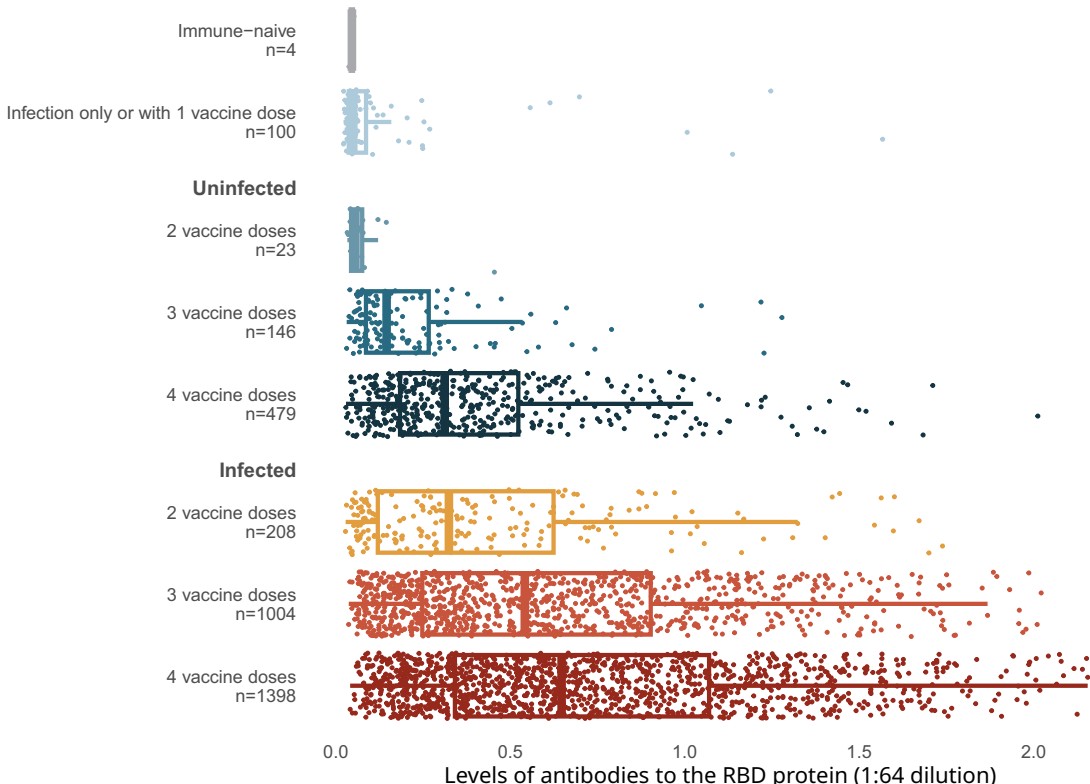

**Appendix 2—figure 3.** Levels of antibodies to RBD stratified by infection and number of vaccination doses.

(A) Uninfected by time of last dose

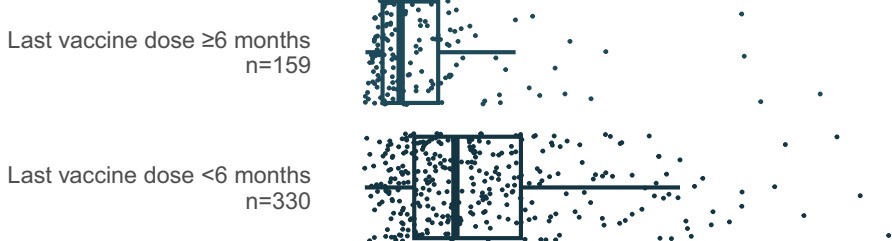

(B) Infected by time of last dose and infection

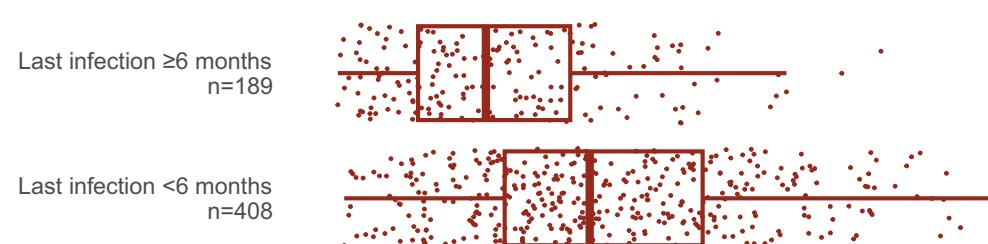

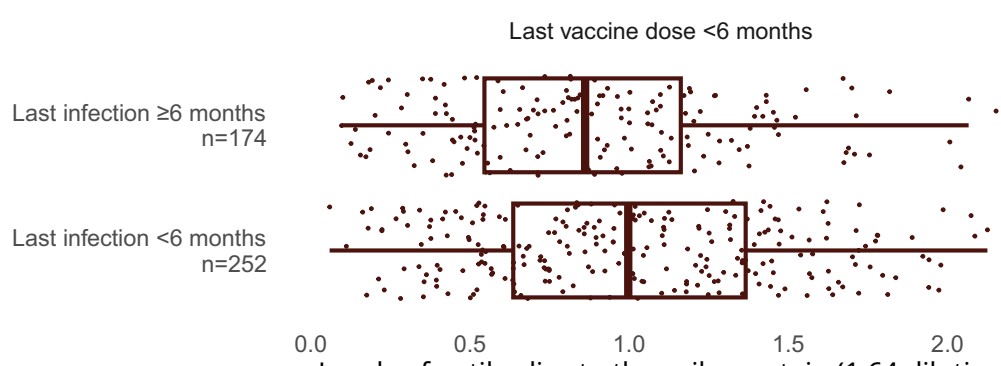

**Appendix 2—figure 4.** Levels of antibodies to the spike protein stratified by infection, vaccination doses, and time since last vaccination or since last infection.

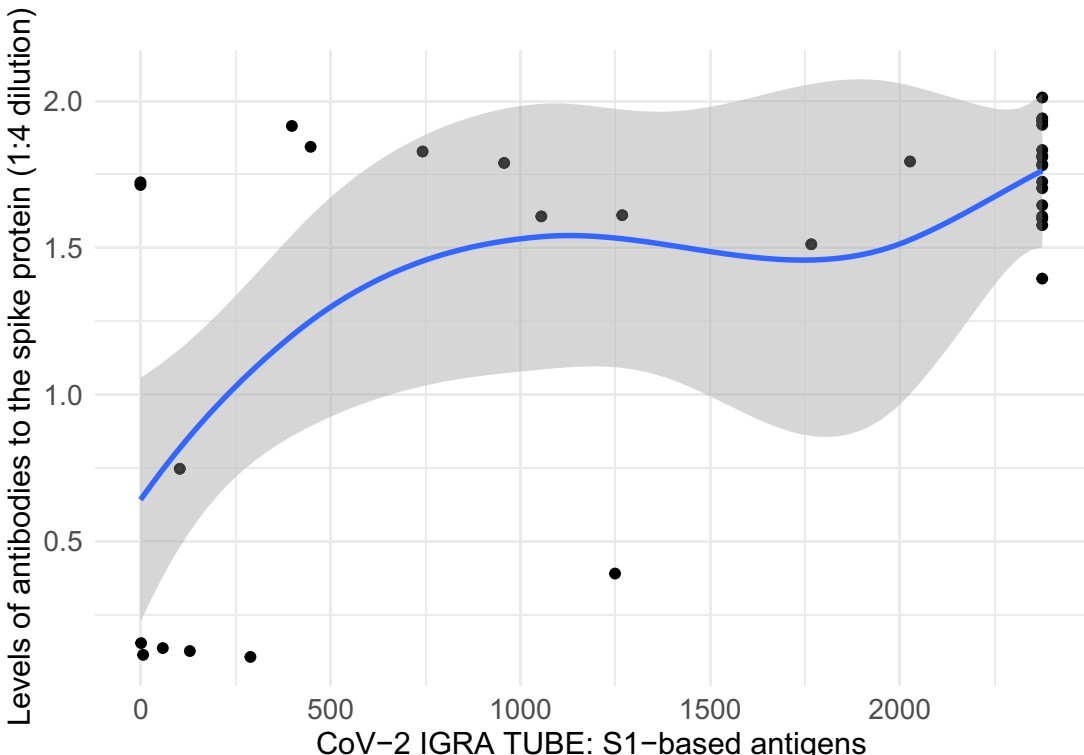

**Appendix 2—figure 5.** Correlation between levels of antibodies to the spike protein and T-cell spike titers. Notes: *x*-axis represents interferon-gamma stimulation on blood samples using antigens based on the SARS-CoV-2 spike protein (https://www.coronavirus-diagnostics.com/documents/Indications/Infections/Coronavirus/ET_2606_D_UK_A.pdf), and *y*-axis represents SARS-CoV-2 spike protein antibodies in dried-blood spot samples. The two variables had a Spearman correlation of 0.508. Smoothed curves and 95% confidence intervals were obtained using locally weighted scatterplot smoothing with span parameter of 0.8 (*Fox and Weisberg, 2018*).

# Appendix 3

**Appendix 3—table 1.** Sample characteristics and representativeness of Phases 4 and 6 for online surveys and DBS samples.

| | 2016 Canadian Census or national surveys (%) | Phase 4 survey | | Phase 4 DBS sample | | Phase 6 survey | | Phase 6 DBS sample | |
|---|---|---|---|---|---|---|---|---|---|
| | | n | % | n | % | n | % | n | % |
| Total (N) | | 14,224 | | 5031 | | 10,088 | | 3378 | |
| High risk regions* | | 4824 | 33.6 | 1732 | 34.0 | 3530 | 34.6 | 1164 | 33.5 |
| Province | | | | | | | | | |
| Ontario | 38 | 5707 | 40.0 | 2103 | 41.5 | 3957 | 39.3 | 1417 | 41.6 |
| British Columbia & Yukon | 14 | 2862 | 20.1 | 1035 | 20.9 | 2120 | 21.1 | 732 | 22.4 |
| Quebec | 23 | 1764 | 12.5 | 617 | 12.2 | 1223 | 12.0 | 364 | 10.5 |
| Prairie provinces & NWT | 19 | 2992 | 21.1 | 979 | 19.6 | 2155 | 21.5 | 653 | 19.4 |
| Atlantic provinces | 7 | 899 | 6.2 | 297 | 5.9 | 633 | 6.2 | 212 | 6.2 |
| Sex | | | | | | | | | |
| Male | 49 | 6453 | 45.9 | 1997 | 40.1 | 4492 | 45.0 | 1271 | 37.8 |
| Female | 51 | 7628 | 53.1 | 3003 | 59.3 | 5515 | 54.2 | 2089 | 61.7 |
| Prefer to self-describe | | 143 | 1.0 | 31 | 0.6 | 81 | 0.8 | 18 | 0.6 |
| Age group (years) | | | | | | | | | |
| 18–39 | 49 | 3632 | 23.9 | 1060 | 19.5 | 2084 | 19.1 | 512 | 13.8 |
| 40–59 | 28 | 5195 | 36.1 | 1752 | 34.4 | 3699 | 35.9 | 1134 | 32.7 |
| 60–69 | 12 | 3303 | 24.5 | 1355 | 28.3 | 2484 | 26.1 | 994 | 30.7 |
| 70+ | 11 | 2094 | 15.5 | 864 | 17.9 | 1821 | 18.9 | 738 | 22.8 |
| Education | | | | | | | | | |
| Some college or less | 45 | 3372 | 34.0 | 1050 | 30.6 | 2395 | 34.3 | 702 | 30.7 |
| College graduate | 32 | 4680 | 31.6 | 1621 | 31.5 | 3333 | 32.1 | 1102 | 32.3 |
| University graduate | 23 | 6172 | 34.4 | 2360 | 37.9 | 4360 | 33.6 | 1574 | 37.0 |
| Visible minority | 22 | 3438 | 23.5 | 820 | 15.8 | 2482 | 24.2 | 525 | 15.3 |
| Indigenous | 5 | 1504 | 11.0 | 495 | 10.2 | 803 | 8.6 | 234 | 7.3 |
| Household size | | | | | | | | | |
| Live alone | 28 | 2614 | 18.7 | 990 | 19.7 | 2070 | 20.8 | 730 | 21.5 |
| Two people | 34 | 6144 | 44.0 | 2328 | 47.0 | 4475 | 45.2 | 1635 | 49.4 |
| Three people | 15 | 2338 | 16.2 | 747 | 14.8 | 1558 | 15.2 | 477 | 14.0 |
| Four people or more | 22 | 3128 | 21.2 | 966 | 18.5 | 1985 | 18.8 | 536 | 15.2 |
| Ever smoking | 54 | 6652 | 49.8 | 2331 | 49.1 | 4781 | 50.6 | 1599 | 50.4 |
| Obesity (≥30 kg/m$^2$) | 27 | 3750 | 27.4 | 1368 | 28.1 | 2675 | 27.7 | 908 | 27.8 |
| Diabetic history | 9 | 1418 | 10.6 | 518 | 11.0 | 1037 | 10.9 | 359 | 11.4 |
| Hypertension history | 23 | 3826 | 28.4 | 1452 | 30.4 | 2850 | 29.7 | 1006 | 31.3 |
| Vaccination[†] | | | | | | | | | |
| Unvaccinated | 10 | 1275 | 9.7 | 209 | 4.5 | 739 | 7.9 | 95 | 3.1 |

*Appendix 3—table 1 Continued*

| | 2016 Canadian Census or national surveys (%) | Phase 4 survey | | Phase 4 DBS sample | | Phase 6 survey | | Phase 6 DBS sample | |
|---|---|---|---|---|---|---|---|---|---|
| | | *n* | % | *n* | % | *n* | % | *n* | % |
| Vaccinated | 90 | 12,949 | 90.3 | 4819 | 95.5 | 9349 | 92.1 | 3283 | 96.9 |
| One dose | 1 | 136 | 1.1 | 23 | 0.5 | 76 | 0.8 | 10 | 0.3 |
| Two doses | 29 | 3807 | 29.9 | 876 | 18.5 | 1246 | 13.0 | 233 | 7.2 |
| Three doses | 32 | 9006 | 69.0 | 3920 | 81.0 | 3801 | 37.2 | 1158 | 34.0 |
| Four or more doses | 28 | | | | | 4226 | 41.1 | 1882 | 55.5 |

*17 high-burden regions identified from a regression analysis of SARS-CoV-2 case counts.

†As of 1 January 2023.

# Appendix 4

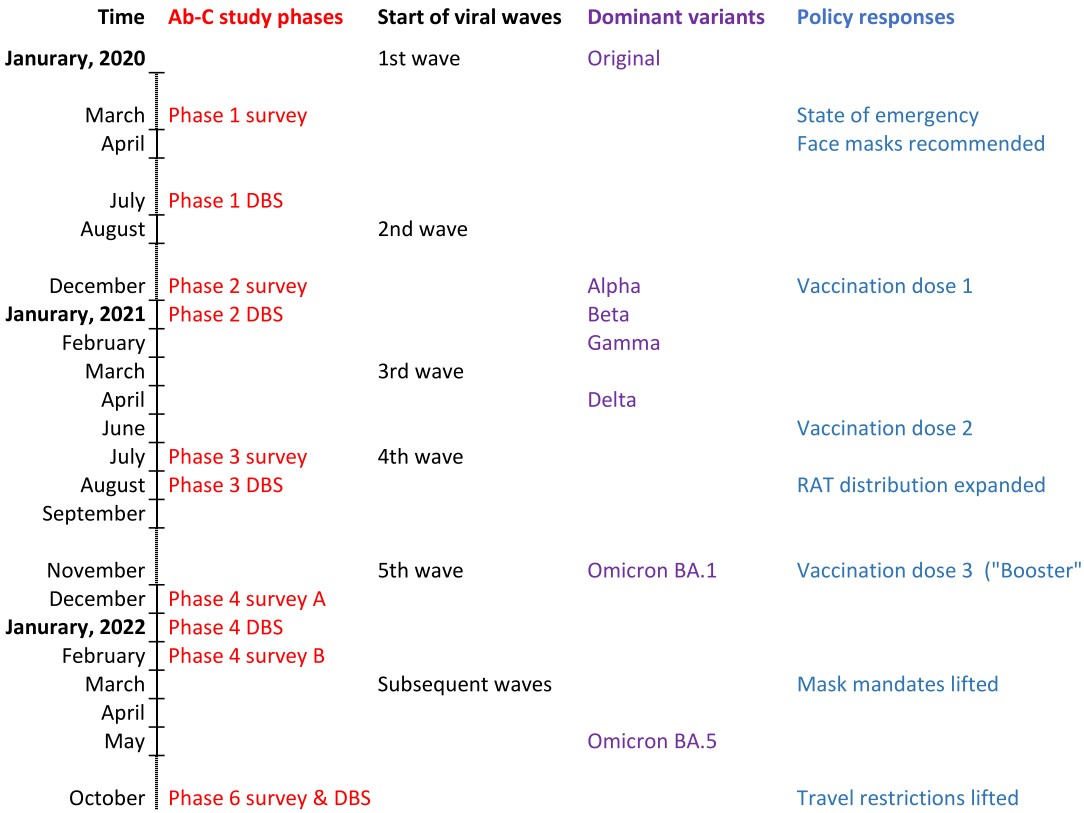

| Time | Ab-C study phases | Start of viral waves | Dominant variants | Policy responses |
|---|---|---|---|---|
| Janurary, 2020 | | 1st wave | Original | |
| March | Phase 1 survey | | | State of emergency |
| April | | | | Face masks recommended |
| July | Phase 1 DBS | | | |
| August | | 2nd wave | | |
| December | Phase 2 survey | | Alpha | Vaccination dose 1 |
| Janurary, 2021 | Phase 2 DBS | | Beta | |
| February | | | Gamma | |
| March | | 3rd wave | | |
| April | | | Delta | |
| June | | | | Vaccination dose 2 |
| July | Phase 3 survey | 4th wave | | |
| August | Phase 3 DBS | | | RAT distribution expanded |
| September | | | | |
| November | | 5th wave | Omicron BA.1 | Vaccination dose 3 ("Booster") |
| December | Phase 4 survey A | | | |
| Janurary, 2022 | Phase 4 DBS | | | |
| February | Phase 4 survey B | | | |
| March | | Subsequent waves | | Mask mandates lifted |
| April | | | | |
| May | | | Omicron BA.5 | |
| October | Phase 6 survey & DBS | | | Travel restrictions lifted |

**Appendix 4—figure 1.** Ab-C study timeline with pandemic waves, dominant variants and policy responses in Canada.

