## [Editor Report]

This study assessed antibody levels, which are indicative of protection, resulting from both COVID-19 vaccination and natural infection in a representative sample of the Canadian population. The work provides solid evidence that Individuals who received a booster vaccination and had a prior infection had the highest antibody levels, particularly when either the vaccination or natural infection had occurred within the past six months. These findings are of fundamental importance in supporting the value of booster vaccination in populations vulnerable to severe COVID-19.

---

## [Decision Letter]

**Decision letter after peer review:**

Thank you for submitting your article "Hybrid immunity from SARS-CoV-2 infection and vaccination in Canadian adults: cohort study" for consideration by *eLife*. Your article has been reviewed by 2 peer reviewers, and I personally oversaw the evaluation in my dual role of Reviewing Editor and Senior Editor.

*Reviewer #1 (Recommendations for the authors):*

Thank you for the opportunity to review this manuscript. This study holds significant importance as it assessed antibody levels arising from both COVID-19 vaccination and natural infection in a representative population-based sample. The analysis was conducted with thoughtfulness and rigor. The sampling methodology ensured the representation of the broader Canadian population, including minorities and indigenous communities. Findings suggest, that despite a substantial number of individuals having been previously infected, especially following the first omicron wave, repeat booster vaccination is essential to ensure that individuals develop an optimal antibody response against new exposures to infection, given the waning of antibodies over time. The study findings carry global significance as it informs decisions about the relevance of booster vaccination for reducing infection incidence amid the ongoing challenge of vaccine hesitancy and the continual emergence of new variants. I have included below a few suggestions that, in my opinion, could enhance the study's contribution to the literature:

1) In reference to the statement "hybrid immunity played a role in reducing severe disease and deaths", I recommend that authors use a more nuanced statement. This is because previous studies have demonstrated that the effectiveness of even only primary-series vaccination against COVID-19 severe disease was high, with slow waning over time. Similar protection was found among unvaccinated individuals who were previously infected with COVID-19. These studies include reference 3 by the authors; Chemaitelly H, Nagelkerke N, Ayoub HH, et al. Duration of immune protection of SARS-CoV-2 natural infection against reinfection. J Travel Med 2022 Sep 30:taac109. doi: 10.1093/jtm/taac109.; Chemaitelly H, Tang P, Hasan MR, et al. Waning of BNT162b2 Vaccine Protection against SARS-CoV-2 Infection in Qatar. N Engl J Med 2021;385:e83.

2) Following on the previous point, I suggest stating explicitly that one objective of repeat booster vaccination is to impart a robust level of protection against acquiring infections.

3) It would be interesting to investigate potential variations in spike protein levels based on the last type of vaccine administered.

4) I recommend that authors comment on the generalizability of the findings considering that individuals who participated in the research may have been different from those who did not participate, and therefore residual confounding cannot be eliminated.

*Reviewer #2 (Recommendations for the authors):*

1. Consider diversifying the sampling strategy to ensure greater representativeness of the Canadian population. Collaborate with healthcare institutions or community organizations to reach demographics underrepresented in online polling platforms. This will enhance the study's external validity and generalizability of findings.

2. Provide comprehensive details on inclusion and exclusion criteria for participant selection, data collection methods, and quality control measures. Transparent reporting of methodology will facilitate reproducibility and enable readers to assess the validity of study findings.

3. Conduct additional validation studies to assess the reliability and accuracy of antibody assays used in the study. Address potential limitations, such as assay saturation, and explore alternative methodologies to enhance the robustness of antibody measurements.

4. Implement strategies to mitigate potential biases associated with self-reported vaccination history and infection status. Consider cross-referencing participant responses with healthcare records or conducting follow-up assessments to validate self-reported data.

5. Exercise caution in interpreting the causal relationship between hybrid immunity and reduced COVID-19 mortality. Acknowledge the multifactorial nature of disease outcomes and consider alternative explanations for observed trends, such as changes in healthcare practices or population behavior.

6. Ensure clarity and coherence in presenting study findings, including graphical representations and statistical analyses. Use clear and concise language to facilitate comprehension and interpretation of results by readers from diverse backgrounds.

7. Address ethical considerations related to participant privacy and informed consent. Provide assurances regarding data confidentiality and adherence to ethical standards in conducting research involving human subjects.

8. Identify areas for future research, such as longitudinal studies to assess long-term immunity dynamics and comparative analyses across different populations. Engage in interdisciplinary collaborations to explore the broader implications of hybrid immunity on public health policy and practice.

---

## [Author Response]

Reviewer #1 (Recommendations for the authors):Thank you for the opportunity to review this manuscript. This study holds significant importance as it assessed antibody levels arising from both COVID-19 vaccination and natural infection in a representative population-based sample. The analysis was conducted with thoughtfulness and rigor. The sampling methodology ensured the representation of the broader Canadian population, including minorities and indigenous communities. Findings suggest, that despite a substantial number of individuals having been previously infected, especially following the first omicron wave, repeat booster vaccination is essential to ensure that individuals develop an optimal antibody response against new exposures to infection, given the waning of antibodies over time. The study findings carry global significance as it informs decisions about the relevance of booster vaccination for reducing infection incidence amid the ongoing challenge of vaccine hesitancy and the continual emergence of new variants. I have included below a few suggestions that, in my opinion, could enhance the study's contribution to the literature:1) In reference to the statement "hybrid immunity played a role in reducing severe disease and deaths", I recommend that authors use a more nuanced statement. This is because previous studies have demonstrated that the effectiveness of even only primary-series vaccination against COVID-19 severe disease was high, with slow waning over time. Similar protection was found among unvaccinated individuals who were previously infected with COVID-19. These studies include reference 3 by the authors; Chemaitelly H, Nagelkerke N, Ayoub HH, et al. Duration of immune protection of SARS-CoV-2 natural infection against reinfection. J Travel Med 2022 Sep 30:taac109. doi: 10.1093/jtm/taac109.; Chemaitelly H, Tang P, Hasan MR, et al. Waning of BNT162b2 Vaccine Protection against SARS-CoV-2 Infection in Qatar. N Engl J Med 2021;385:e83.

We have edited the manuscript to reflect a more nuanced phrasing. We have added these references.

2) Following on the previous point, I suggest stating explicitly that one objective of repeat booster vaccination is to impart a robust level of protection against acquiring infections.

We have edited the text accordingly in the Discussion section.

3) It would be interesting to investigate potential variations in spike protein levels based on the last type of vaccine administered.

Since Phase 2 of our Action to Beat Coronavirus (Ab-C) study, we have asked our participants about the time and name (Pfizer/Moderna/AstraZeneca/other) of each COVID vaccine they received. However, as we merged data from Phase 2 to Phase 6 (the data used for this study), we found inconsistencies in the self-reported name of COVID vaccines. Some participants reported different vaccine names for the vaccines they had reported on previously. Because we had no way of confirming which iteration of the information was correct, we could not distinguish recall bias from genuinely correcting previously erroneous responses. Thus, we decided not to use such information in the analyses.

4) I recommend that authors comment on the generalizability of the findings considering that individuals who participated in the research may have been different from those who did not participate, and therefore residual confounding cannot be eliminated.

We have edited the Discussion section accordingly.

Reviewer #2 (Recommendations for the authors):1. Consider diversifying the sampling strategy to ensure greater representativeness of the Canadian population. Collaborate with healthcare institutions or community organizations to reach demographics underrepresented in online polling platforms. This will enhance the study's external validity and generalizability of findings.

We acknowledge that our online panelists generally have higher education attainment than the average Canadian population. Other than education, our panelists’ demographic distributions are similar to the Canadian census. We oversampled the elderly and the indigenous groups to counter under-responsiveness from those groups. Importantly, these selection biases did not change appreciably during the various study phases, and given vaccination and infection (particularly from the Omicron variant) were widespread, the interpretation of trends should not be biased. We mention this point on page 4.

2. Provide comprehensive details on inclusion and exclusion criteria for participant selection, data collection methods, and quality control measures. Transparent reporting of methodology will facilitate reproducibility and enable readers to assess the validity of study findings.

We have revised the methods section to clarify inclusion and exclusion criteria.

3. Conduct additional validation studies to assess the reliability and accuracy of antibody assays used in the study. Address potential limitations, such as assay saturation, and explore alternative methodologies to enhance the robustness of antibody measurements.

We acknowledge that assay validations are important and warrant diligent research by itself. We have already included in the appendix the details of the validation steps taken and the generally high performance (high sensitivity and specificity) of the chemiluminescent assay.

4. Implement strategies to mitigate potential biases associated with self-reported vaccination history and infection status. Consider cross-referencing participant responses with healthcare records or conducting follow-up assessments to validate self-reported data.

While we have permission from participants to link to their health care records, the linkage process takes considerable time in Canada. However, other evaluations such as those done by the Toronto COVID database show reasonably good concordance of self-reported symptoms (https://www.ncbi.nlm.nih.gov/pmc/articles/PMC8730782/).

5. Exercise caution in interpreting the causal relationship between hybrid immunity and reduced COVID-19 mortality. Acknowledge the multifactorial nature of disease outcomes and consider alternative explanations for observed trends, such as changes in healthcare practices or population behavior.

We have revised the text to reflect caution when interpreting causal relationship between hybrid immunity and reduced COVID-19 mortality, as well as addressing the multiple factors related to disease outcomes.

6. Ensure clarity and coherence in presenting study findings, including graphical representations and statistical analyses. Use clear and concise language to facilitate comprehension and interpretation of results by readers from diverse backgrounds.

We have revised our methods and Discussion sections accordingly.

7. Address ethical considerations related to participant privacy and informed consent. Provide assurances regarding data confidentiality and adherence to ethical standards in conducting research involving human subjects.

The Ab-C received full IRB review at Unity Health Toronto to address these concerns. We now note that we obtained informed consent from the participants. Data are stored securely, separately for identifiable information and study data. All aspects of this study adhere to ethical standards of research involving human subjects.

8. Identify areas for future research, such as longitudinal studies to assess long-term immunity dynamics and comparative analyses across different populations. Engage in interdisciplinary collaborations to explore the broader implications of hybrid immunity on public health policy and practice.

We have revised the text to include directions of future research.